# Imidazolium Ionic Liquids as Compatibilizer Agents for Microcrystalline Cellulose/Epoxy Composites

**DOI:** 10.3390/polym15020333

**Published:** 2023-01-09

**Authors:** Eduardo Fischer Kerche, Agnė Kairytė, Sylwia Członka, Vinícius Demétrio da Silva, Nicholas Alves Salles, Henri Stephan Schrekker, Sandro Campos Amico

**Affiliations:** 1Programa de Pós-Graduação em Engenharia de Minas, Metalúrgica e de Materiais (PPGE3M), Federal University of Rio Grande do Sul (UFRGS), Porto Alegre, RS 91501-970, Brazil; 2Laboratory of Thermal Insulating Materials and Acoustics, Faculty of Civil Engineering, Institute of Building Materials, Vilnius Gediminas Technical University, LT-08217 Vilnius, Lithuania; 3Institute of Polymer & Dye Technology, Lodz University of Technology, 90-924 Lodz, Poland; 4Laboratory of Technological Processes and Catalysis, Federal University of Rio Grande do Sul (UFRGS), Porto Alegre, RS 91501-970, Brazil

**Keywords:** imidazolium salt, hydrogen bonding, DGEBA/TETA epoxy, surface treatment, biobased composite, mechanical behavior

## Abstract

Four imidazolium-based ionic liquids (IL; 1-butyl-3-methylimidazolium chloride, 1-carboxymethyl-3-methylimidazolium chloride, 1,3-dicarboxymethylimidazolium chloride and 1-(2-hydroxyethyl) -3-methylimidazolium chloride) were tested as compatibilizers of microcrystalline cellulose (MCC). Subsequently, ethanolic IL solutions were prepared; MCC was mixed, and the mixtures were left to evaporate the ethanol at ambient conditions. These modified MCC were characterized and applied as reinforcements (5.0 and 10 phr) in an epoxy resin aiming to manufacture biobased composites with enhanced performances. The IL did not significantly modify the morphological and structural characteristics of such reinforcements. Regarding the thermal stability, the slight increase was associated with the MCC-IL affinity. The IL-modified MCC-epoxy composites presented improved mechanical responses, such as flexural strength (≈22.5%) and toughness behavior (≈18.6%), compared with pure epoxy. Such improvement was also obtained for the viscoelastic response, where the storage modulus at the glassy state depended on the MCC amount and IL type. These differences were associated with stronger hydrogen bonding between IL and epoxy hardener or the IL with MCC, causing a “bridging” effect between MCC and epoxy matrix.

## 1. Introduction

Epoxy resin is a thermoset polymer designated for advanced applications. The aerospace sector, paints, coatings, construction, electronics, adhesives and composite materials are some examples of epoxy resin application, with a considerable market in the last decade [1]. Composites based on diglycidyl ether of bisphenol A (DGEBA) present some inherent characteristics, such as high mechanical properties, thermal stability and harsh environment resistance [2].

Although the aforementioned benefits of the fabrication of composites using epoxy resin as matrix exist, there are still some limitations to overcome. The main drawback of these resins is their brittle characteristics, due to the high content of aromatic rings in their chemical formulations forming highly rigid 3-D crosslinked structures [3]. Some strategies have been pursued to bypass this fragility, which include increasing the chain length between crosslinks by using new technologies such as hierarchical curing processes [4] and varying the amount of curing agent, which aims to induce a second phase into the rubber [5].

Simple strategies may also be employed for the improvement of the mechanical response of epoxy and its composites, such as the use of fillers. Some examples include the use of inorganic particles, such as nano clays, cork particles [6,7,8] and nanoparticles, such as carbon nanotubes [9], graphene and its derivatives [10], and nanocellulose [11].

Cellulose is the most abundant biopolymer in the world and is responsible for the high strength and stiffness of plants; its fibers are high potential reinforcements for the manufacturing of environmentally friendly composites [12]. Microcrystalline cellulose (MCC) is one such filler example used to enhance mechanical, dynamical mechanical and thermal properties of composites [13]. The production of renewable MCC is related to carbon footprint reduction and, as a consequence, linked to the manufacturing of products with a high environmental appeal. For these reasons, strategies have been developed for the use of cellulose as reinforcement or dispersed phase in polymers, aiming to improve their physical-mechanical characteristics [14].

Regarding the preparation of MCC-reinforced epoxy-based composites, Neves et al. [15] increased the thermal stability, the storage moduli in both the glassy (119%) and rubbery states (127%), and the loss modulus (173%), compared to the neat resin, even when a low content of 2.5 wt.% of MCC was employed. Despite these improvements, due to the high content of cellulose hydroxyls (especially those linked to C_2_, C_3_ and C_6_) [16,17], many hydrogen bonds between adjacent chains are formed, which favor the aggregation of MCC, decreasing the dispersion of such reinforcement. In addition, if a poor fillers’ dispersion is achieved, the formed aggregates may induce a premature and undesirable failure, reducing the overall composite performance [18,19].

Indeed, a better dispersion of fillers contributes to the enhancement of physical-mechanical properties of composites [20,21]. There are several chemicals for the improvement of interaction of fillers with epoxy resin that use chemical reagents, such as hydrochloric acid, nitric acid, sulfuric acid, acetone, tetrahydrofuran, thionyl chloride, and ethylenediamine, aiming to insert compatible groups on the filler surface [9,22]. Specifically, for cellulose composites and their derivatives, Yue et al. [23] increased close to 670 times the stiffness of an epoxy-based composite with 10 wt.% of an amino trimethoxy silane-modified cellulose nanocrystal, which was ascribed to the better dispersion of this filler.

Imidazolium ionic liquids (IL), especially those physisorbed, may be used to modify the interfacial characteristics of fillers for the manufacturing of composites with improved properties [24]. Simultaneously, such IL treatment of polyaramid pulp increased the surface roughness and favored the manufacturing of enhanced rubber composites [25,26,27], filled-polyurethane foams [28], and epoxy-based composites [8,27]. Although, in most cases, IL does not form covalent bonds with the filler or the matrix, the non-covalent interactions through the imidazolium’s cation π-cloud in π–π stacking and hydrogen atoms (C_2_–H, C_4_–H and C_5_–H) in hydrogen bonds make these compounds great candidates for the compatibilization of reinforcement into composite materials.

Many studies have been reported about the disruption of secondary bonds formed between fillers with low dimensions when IL are used, even at low contents [29]. Multiwall carbon nanotubes and graphene nanoplatelets are some examples of fillers modified by IL which aim to produce epoxy-based composites with improved mechanical, electrical/thermal conductivity/stability, and tensile strength due to the better fillers’ dispersion [18,30,31].

Despite the aforementioned studies regarding the use of IL for dispersion of nano and micro reinforcements into epoxy matrix, to our knowledge, there are no studies about the incorporation of a modified MCC with IL for the manufacturing of enhanced composites. As also demonstrated, the use of IL may be an interesting alternative, since the components are used in lower contents compared, for example, with other chemicals, such as silane coupling agents.

Within this panorama, and considering the relatively high cost of the cited chemicals, their use in lower contents would favor the development of economically viable MCC-reinforced, epoxy-based composites with improved physical-mechanical behavior. Hence, the objective of this work was to treat MCC with 1 wt.% of IL, testing the four IL presented in Figure 1 and studying the effect on the reinforcement of epoxy resin, which was investigated through mechanical and dynamical mechanical properties. Different cations were studied, since in the previous report IL with chloride anion presented the greater compatibility with the epoxy system and improved the response of such composites [8].

## 2. Materials and Methods

Commercial epoxy resin DGEBA AR260^®^ (complex viscosity 1350 mPa.s and density 1.16 g/cm^3^ at 25 °C) and triethylenetetramine (TETA) hardener AH260^®^ (complex viscosity 220 mPa.s and density 0.98 g/cm^3^ at 25 °C) were purchased from Barracuda Co. (São Paulo, Brazil). MCC (diameter of ca. 20 µm, density 1.5 g/cm^3^, ignition temperature 232 °C, pH 7.0002 (100 g/L, H₂O, 20 °C)) and the IL 1-butyl-3-methylimidazolium chloride (C_4_MImCl, Figure 1) were acquired from Sigma-Aldrich (St. Louis, MO, USA). All commercial chemicals were used as received. Literature procedures were used to synthesize 1-carboxymethyl-3-methylimidazolium chloride [32], 1,3-dicarboxymethylimidazolium chloride [32], and 1-(2-hydroxyethyl)-3-methylimidazolium chloride [33] (HO_2_CC_1_MImCl, (HO_2_CC_1_)_2_MImCl and C_2_OHMImCl, respectively, Figure 1) [33,34,35].

The ^1^H NMR spectra of each IL and IL-TETA mixtures were recorded in a Bruker (Billerica, MA, USA) (400 MHz) equipment, model Avance I, at room temperature (see in the Appendix A). The chemical shifts are given in ppm and referenced to the residual solvent signal (DMSO-*d*6 = 2.50 (^1^H)). Thermogravimetric analysis (TGA) was performed in a TA Instruments Q50 thermogravimetric analyzer (New Castle, DE, USA), using an average sample weight of 10 mg of the cited IL, a heating rate of 20 °C/min, from 50 to 450 °C, and a N_2_ atmosphere.

### 2.1. MCC Treatment and Characterization

MCC treatment was performed using a reported procedure [29]. Briefly, ethanolic solutions with 1 wt.% of IL in relation to the MCC mass were prepared, and the MCC was added. These mixtures were placed in an ultrasonic bath (50–60 Hz) at 50 °C for 30 min, left to evaporate the solvent for 48 h at room temperature and then placed in a vacuum oven for 12 h at 60 °C. It is important to stress that this amount of IL was chosen due to the previous report, where an amount of 4 wt.% of IL or higher caused undesirable changes in thermal, morphological and structural characteristics of MCC [29].

X-ray powder diffraction (XRD) was assessed in a PanAlytical MDP Shimadzu XRF1800 equipment at 40 kV and 17 mA with monochromatic CuK, λ = 0.1542 nm. The intensities were measured in the 5° > 2θ < 40° range (0.05°/4 s) and the crystalline index (I_c_) was calculated following the Segal method [23]. Field emission scanning electron microscopy (FEG-SEM) and energy dispersive X-ray spectroscopy (EDS) were performed in a Jeol JSM 6060 SEM equipment, using 10 kV of voltage. TGA was performed using the same equipment and methodology aforementioned, and Fourier-transform infrared spectroscopy (FTIR) was performed in the mid-infrared range (4000–500 cm^−1^) with the resolution of 4 cm^−1^, using a Nicolet 6700 ATR-IR equipment, from Thermo Fisher Scientific (Waltham, MA, USA).

### 2.2. Epoxy Composites Preparation and Characterization

The commercial MCC or IL-treated MCC were added to the epoxy resin followed by 3 h of heated (70 °C) mechanical stirring (~400 rpm). The mixtures were placed in an ultrasonic bath for 20 min at 50–60 Hz; the TETA hardener (26.0 phr) was added, and the mixtures were, again, mechanically stirred (now at room temperature) for 10 min. Afterwards, the mixture was poured into silicone molds and subjected to three cycles of negative pressure (−0.8 bar; 10 min on negative pressure and 10 min under atmospheric air) to remove bubbles, and then the composites were cured at room temperature for 24 h. Subsequently, the specimens went through a post-curing process in an oven at 60 °C for 16 h, according to the manufacturer’s protocol [36].

The adopted nomenclature uses E for epoxy, MCC for microcrystalline cellulose, and MCC_IL for MCC treated with the corresponding IL (C_4_ = C_4_MImCl; C_2_OH = C_2_OHMImCl; HO_2_ = HO_2_CC_1_MImCl; (HO_2_)_2_ = (HO_2_CC_1_)_2_MImCl). The numbers 5 and 10 for, respectively, 5 and 10 parts per hundred (phr) of MCC in relation to the resin.

Differential scanning calorimetry (DSC) was performed with just-prepared, uncured composites in a DSC Q2000 equipment from TA Instruments (New Castle, DE, USA) in the temperature range of −80–250 °C at a heating rate of 10 °C/min. The evaluated thermodynamic parameters were the curing onset temperature (T_onset_), maximum curing rate temperature (T_peak_), and total heat of reaction (ΔH), the latter calculated by the integration of the exothermic peak caused by the resin cure.

Tensile strength (TS), tensile modulus (E_1_), Poisson ratio (ν) and shear modulus (G) were determined according to ASTM D638–14. Dumbbell-shaped specimens (length of 165 mm, thickness of 3.2 mm and at the useful space width of 16 mm) were used for the test, which was performed in a universal test machine, Instron^®^ 3382 (Norwood, MA, USA), equipped with a load cell of 5 kN, and test speed of 5.0 mm/min. Clip-gauge was used to assess the strains in directions parallel and perpendicular to the load application.

Flexural strength (σ_f_) and flexural modulus (E_f_) were obtained following ASTM D790–17, using 70 × 12.7 × 3.0 mm specimens. Three-point bending tests were performed using an Emic^®^ universal testing machine, model 23-5D (Norwood, MA, USA), equipped with a 5 kN load cell, and test speed of 1.28 mm/min.

Fracture toughness was assessed, according to ASTM D5045–14, using single-edge notch bending (SENB) specimens with 75 × 10 × 5 mm dimensions. The notch was made with a Ceast NotchVis V-notch (Beijing, China) machine in the middle of the specimen’s length, and a pre-crack was made until reaching half the width by carefully sliding a sharp razorblade. After testing, the cracked sample’s surface was analyzed via optical microscopy prior to the calculations of the critical stress intensity factor (K_IC_) (Equation (1)) and the critical strain energy release rate (G_IC_) (Equation (3)). The length of the zone of plastic deformation (ZPD) was measured using SEM images obtained with a Phenom microscope (Beijing, China), model ProX desktop, at a voltage of 10 kV. At least three samples were tested to validate the results.
(1)$$K_{\mathrm{IC}}=\left(\frac{P_{Q}}{{\mathrm{WB}}^{1/2}}\right)f\left(x\right)$$
(2)f(x)=6x12×[1.99 − x(1 − x)(2.15 − 3.93x+2.7x2)](1+2x)(1 − x)32
(3)$$G_{\mathrm{IC}}=\frac{U}{\left(\mathrm{WB}\varphi \right)}$$
(4)φ=α+18.64∂α∂x
(5)α=[16x2(1 − x)2][8.9 − 33.717x+79.616x2 − 112.952x3+84.815x4 − 25.672x5]
where: P_Q_ is the maximum supported load by the sample; W and B are the sample’s width and depth, respectively; f(x) is a function that depends on the crack geometry after testing, defined by Equation (2); U is the energy generated by the applied strain, obtained by integrating the curve of load vs. displacement; φ and α are factors to calibrate the measurements, defined by Equations (4) and (5), respectively; and x is the ratio between the crack’s length and total specimen width.

At least five specimens were evaluated for each mechanical test, as suggested by each standard. Inferential statistical analysis was performed on the experimental mechanical data. Normality and homogeneity of each level were verified by Shapiro–Wilk and Levane tests, respectively. After that, one-way ANOVA and averaging tests were performed following the LSD Fischer procedure with 90% confidence.

Dynamical mechanical analyses (DMA) were performed with prismatic specimens (dimensions: 40.0 × 12.7 × 3.0 mm) and three-point bending clamps in a TA Instruments DMA 2980 equipment (New Castle, DE, USA) at a frequency of 1.0 Hz, from 30 to 200 °C. The T_g_ was determined from the tan delta peak. The effectiveness of filler reinforcement constant, C (Equation (6)), was calculated, aiming to evaluate the influence of the content of MCC and the used IL for the surface modification, as suggested by Ornaghi et al. [37]:(6)$$C=\left(\frac{{\left({E’}_{g}{/E’}_{r}\right)}_{\mathrm{Composite}}}{{\left({E’}_{g}{/E’}_{r}\right)}_{\mathrm{Resin}}}\right)$$
where: E′_g_ and E′_r_ are the storage moduli related to the glassy and rubbery regions, respectively.

## 3. Results and Discussion

### 3.1. MCC Surface Treatment

Figure 2 presents the XRD diffraction patterns of MCC (unmodified) and IL.Cl (1 wt.% IL-modified MCC). The crystalline peaks, located at 14.8–16.5° and 22.4°, are related to the crystalline part of MCC and correspond to the (110) and (200) crystallographic planes, from cellulose I, respectively. The amorphous halo, between 16.5–20°, was also observed for all IL.Cl [17]. The crystallinity indexes calculated for MCC; MCC_C_4_; MCC_HO_2_; MCC_(HO_2_)_2_ and MCC_C_2_OH were 86.2%, 88.5%, 87.5%, 87.7% and 90.2%, respectively. An increase in the crystallinity was observed for all IL-modified MCC, highlighting MCC treated with C_2_OHMImCl. In general, the stronger acidic hydrogen (C_2_–H) of the imidazolium ring, present in all tested IL, strengthens the interaction with MCC through hydrogen bond formation [29]. This behavior may have been more pronounced due to the hydroxyl group of C_2_OHMImCl, which most likely explains the highest crystallinity of MCC_C_2_OH.

The FT-IR spectra (Figure 3) of the IL-treated MCC samples display the characteristic MCC bands, including those at 3050 cm^−1^ (stretching vibration of O—H) and 2793 cm^−1^ (symmetric and asymmetric C—H vibration) [17,38]. The peaks at 1640, 1430, 1314 and 1056 cm^−1^ are related to the O—H bending vibration of absorbed water, the symmetrical bending of CH_2_ groups, the CH_2_ wagging, and the C—O deformation at C_6_ [13], respectively. Although physisorbed IL will not modify the characteristic MCC peaks due to the absence of chemical transformations, their interaction could cause these peaks to shift, but this was not observed and could be a consequence of the low IL content of 1 wt.%. Most likely, this was also the source for the absence of IL characteristic peaks, as observed in other studies [6,25,29].

SEM micrographs and EDS patterns of the studied MCC samples are given in Figure 4. All samples presented irregular rod shapes with a uniform distribution of carbon and oxygen atoms. Although IL were not identified by FTIR for IL-treated MCC, EDS detected the IL-related chloride atoms. The regular chloride patterns correspond to a good distribution of all IL on the MCC surface. Moreover, compared with the other samples, a greater irregularity and apparent disruption of crystals was evidenced for those MCC_HO_2_ and MCC_(HO_2_)_2_., as presented in the circles in Figure 4c,d, respectively. These differences were also found for aramid fiber treated with HO_2_CC_1_MImCl or (HO_2_CC_1_)_2_MImCl, which was related to the acidic carboxylic group(s) of these salts, that caused a greater disruption of hydrogen bonding between the adjacent fillers [39]. MCC_C_2_OH presented lower disruption of crystals, compared to the other MCC_IL, despite being rougher than that of pristine MCC, as evidenced by the arrows in Figure 4e.

Figure 5 shows the TGA results for the MCC samples, and their thermal decomposition data are summarized in Table 1. For cellulosic materials, three main weight-loss events can be assessed: ≈65 °C, related to the evaporation of water; ≈362 °C, attributed to the dehydration of cellulose by an endothermic process followed by thermal depolymerization; and ≈374 °C, related to the thermal decomposition of cellulose into ᴅ-glucopyranose monomers [13,40]. Higher T_max_ values were obtained for all MCC_IL, highlighting the highest T_max_ of 377.3 °C of MCC_C_4_. This could be originated from the slightly increased crystallinities as was determined by XRD, which was reflected in higher thermally stable bonds between adjacent cellulose chains [17]. In other words, the higher the crystallinity of MCC, the poorer the heat diffusion along the sample, improving heat-resistance, which also decreases the T_onset_ values for those MCC_C_4_ and MCC_C_2_OH [29].

Regarding the weight loss at 374 °C, all IL treatments improved the thermal stability of MCC, highlighting again MCC_C_4_. This result is supported by the weight loss at T_max_, where those MCC_C_4_ presented the lower values among all the samples. Finally, Appendix A presents the TG and DTG curves of the studied IL, and their higher thermal stability than that of MCC may have also influenced the overall thermal stability of the MCC_IL samples, together with the IL-promoted higher MCC crystallinity and stronger physical interactions among the crystals, when they are not incorporated into the epoxy composite.

### 3.2. Interaction of IL with Epoxy Matrix

The hydrogen bond formation between all aforementioned IL and the AH260 hardener was studied by ^1^H NMR spectroscopy. The spectra of C_4_MImCl and C_4_MImCl + AH260 (Appendix A), HO_2_CC_1_MImCl and HO_2_CC_1_MImCl + AH260 (Appendix A), (HO_2_CC_1_)_2_MImCl and (HO_2_CC_1_)_2_MImCl + AH260 (Appendix A), and C_2_OHMImCl and C_2_OHMImCl + AH260 (Appendix A) confirmed these interactions. When IL were in the presence of hardener, the peaks related to the imidazolium ring hydrogens (C_2_–H, C_4_–H and C_5_–H) shift to other frequencies, depending on the IL type for higher or lower.

When the hydrogen bond donor-acceptor distance decreases as a result of stronger hydrogen bonding, shielding of the hydrogen bond acceptor becomes smaller [41]. The level of proton de-shielding may be associated with the extension of hydrogen bond strengthening [8]. For the hydrogen from the imidazolium ring (i.e., C_2_–H), the greater displacements were observed, from 9.57 ppm to 9.84 ppm for C_4_MImCl + AH260, from 9.22 to 9.05 for HO_2_CC_1_MImCl + AH260, from 9.23 to 9.01 for (HO_2_CC_1_)_2_MImCl + AH260, and from 9.23 to 9.28 for C_2_OHMImCl + AH260. From the NMR de-shielding evaluation, it is possible to assess that C_4_MImCl formed stronger hydrogen bonding with AH260, followed by C_2_OHMImCl. Moreover, those (HO_2_CC1)_2_MImCl + AH260 and HO_2_CC1MImCl + AH260 presented a lower de-shielding, due to the presence of CO_2_H groups from the carboxylic acid in the molecule.

### 3.3. MCC-IL/Epoxy Composites’ Properties

#### 3.3.1. Curing Kinetic

Appendix A shows the DSC curves for the curing kinetics of epoxy and the MCC/epoxy composites, and Table 2 summarizes the data obtained from these curves. All samples presented an expressive exothermic peak during the cure, indicating high crosslinking densities. Compared to epoxy, the incorporation of MCC, without or with IL, did not cause significant differences in T_onset_ and T_peak_ [42]. On the other hand, when evaluating the heat of reaction, E/MCC10 presented a slight increment in comparison to epoxy.

The use of compounds with free hydroxyls at the surface can considerably increase interaction of epoxy resins with amine hardeners or other nucleophilic agents. Indeed, the carbon of the epoxy ring is susceptible to this attack and, as such, these oxygen functionalities can serve as catalysts, accelerating the curing reaction and increasing the total heat of the reaction [43]. On the other hand, the use of IL slightly decreased the total heat of the reaction, which may be attributed to the steric hindrance promoted by the IL at the MCC surface, which hinders the aforementioned interaction of MCC with epoxy and its hardener.

#### 3.3.2. Tensile, Flexural and Fracture Toughness Behavior

Typical curves of the tensile and flexural tests are presented in Appendix A, respectively, and are similar for all composites, independent of the IL or MCC used. Nevertheless, neat epoxy presented a somewhat different behavior, a greater plastic deformation. This was not evident for the composites, which was probably related to the high content of MCC used as reinforcement [23], highlighted by the flexural test curves.

Table 3 joins the tensile test results, and it is evident that when MCC was treated with some IL, a significant improvement in both tensile and shear modulus occurred for the composites, especially for E/MCC_HO_2_10 and E/MCC_C_2_OH10. This could be due to a better filler distribution, yielding higher load transferring between the polymer matrix and reinforcement [18,24,44,45,46], stiffening the overall composite. This behavior could also be responsible for the decrease in the Poisson’s ratio, when the composites with IL-treated MCC are compared with those with pristine MCC.

The composite’s mechanical properties may be linked with the constituent properties by applying the corresponding principle of viscoelasticity to the elastic problem. Then, the Poisson’s ratio becomes imminent because the composite mechanical response is much more sensitive to the reinforcement stiffness than that of the matrix. Consequently, particles that are better distributed facilitate the load transferring by shear stresses at the interface. This mechanism reduces the deformability in the transversal direction of load application for the tensile tests [47] and, consequently, decreases the values of ν. The IL-MCC composites presented lower values for ν compared to unmodified MCC/epoxy, especially E/MCC_C_2_OH10. Then, IL may increase the load transferred between epoxy matrix and MCC, perhaps by the hydrogen bonding formed between MCC and epoxy hardener as evident in NMR results.

This hypothesis is further supported by the flexural test results summarized in Table 4. The improvements in flexural modulus and strength stress the direct correlation with the use of IL and the content of 10 phr for the reinforcement of epoxy resin, especially for E/MCC_C_4_10, E/MCC_HO_2_10 and E/MCC_C_2_OH10. These IL-enhanced flexural moduli and strengths are an indicative of their effectiveness on the surface treatment of MCC and compatibilization with epoxy. Differently from tensile tests (see Table 3), the flexural behavior of composites is lower depending on the failure type and sample preparation; then, lower standard deviation and better accuracy in the measure of failure is evident.

The results for the toughness properties are given in Table 5. There was a decrease on the K_IC_ for the composites with 5 phr of MCC compared with the neat epoxy, as well as for those E/MCC_(HO_2_)_2_10 and E/MCC_C_2_OH10. This difference may be related to the lower strain at break for both tensile and three-point bending tests (see Table 3 and Table 4), since a lower plastic behavior is related to brittle materials. On the other hand, those E/MCC_C_4_10 and E/MCC_HO_2_10 presented a significant improvement in G_IC_ that may be attributed to the better anchoring of the particles to the epoxy resin, promoted by the hydrogen bonding of the IL with the cured epoxy and a higher crack propagation deviation.

According to previous works [48,49,50], increased toughness of epoxy-based composites can be attributed to the presence of particles as obstacles, which move the crack propagation. Such effect is only observed when the particles present good anchoring by the matrix, which also changes the failure pattern. Then, those samples treated with C_4_MImCl presented increments in both toughness, flexural, and tensile tests, which may be attributed to the stronger hydrogen bonding between the IL and the cured resin.

Figure 6 presents the SEM micrographs of the cracked samples’ surfaces from SENB tests. For epoxy resin (Figure 6a), a smooth and flat fracture surface was obtained due to its fragile behavior. On the other hand, the incorporation of MCC (E/MCC5) caused a rougher fracture surface and slightly increased ZPD (Figure 6b), indicating a greater capacity to absorb energy by plastic deformation [48,51]. Compared to E/MCC5, E/MCC_C_4_5 presented an apparent better dispersion of MCC (as evidenced by the circles in Figure 6b,c, respectively), which may be related to the IL’s compatibilizer effect through the hydrogen bond formed with the epoxy network [8].

Those E/MCC_(HO_2_)_2_10 and E/MCC_C_2_OH10 do not present significant differences in ZPD, compared with E/MCC_C_4_10, indicating that the type of IL does not influence the initiation of fracture. Despite that no significant differences for ZPD were found for the composites, all samples presented a significant difference compared with the neat epoxy, as evidenced in Figure 6a. When IL-MCC is employed, an even greater ZPD is found, which may be related to the greater compatibility of the fillers with the epoxy system.

#### 3.3.3. Viscoelastic Behavior

The storage modulus curves for the epoxy resin and its MCC-IL composites are shown in Figure 7, and the corresponding results are summarized in Table 6. All samples presented the three main regions that are characteristic for thermoset polymers: glassy (about 40–80 °C), glass transition, and rubbery (above 120 °C). Both in the glassy and rubbery regions, E/MCC_C_4_5, E/MCC_(HO_2_)_2_10 and E/MCC_C_2_OH10 afforded improved storage moduli. For the corresponding composite of E/MCC_C_4_5 with the higher MCC_C_4_ content of 10 wt.% (E/MCC_C_4_10), the gains in storage moduli were not only lost, but now the effect of IL was detrimental due to particle-particle touching and a lower interface formed between the epoxy matrix and MCC filler, which decreases the values for the normalized E’ in both glassy and rubbery regions [8,46,52] compared to E/MCC_C_4_5.

Lower C values were also reported for those E/MCC_C_4_5, E/MCC_C_4_10 and E/MCC_HO_2_10. Low C constants correspond to greater filler reinforcement effectiveness. This is the outcome of a complex combination of factors, including the strength of interfacial interaction, filler aggregation and filler dispersion. In general, a higher amount of interface between the composite’s components is mainly responsible for the restriction of polymeric epoxy networks, requiring a higher energy for their vibration and shifting E’_g_ to greater values [42].

The loss modulus (E”) curves are compilated in Figure 8, which correspond to the viscous responses of the materials, measuring the dissipated energy in heat form [42]. Those E/MCC_C_4_5, E/MCC_(HO_2_)_2_10, and E/MCC_C_2_OH presented the higher peak heigh among all samples, which can be related to the greater capacity of such composites dissipating energy in heat form [12]. Furthermore, all composites presented a broader and higher E” peak (compared to neat epoxy) without secondary relaxations (as β or γ transitions), regardless of the IL treatment, which is characteristic for composites with a low content of aggregates [52]. Higher peaks are expected for composite materials compared to neat polymers, since a second phase represents a discontinuity in the system [8]. In addition, broadening of the E” peak is a consequence of material strengthening [53].

E/MCC_C_4_5, E/MCC_(HO_2_)_2_10 and E/MCC_C_2_OH10 presented the broadest and highest E” curves, highlighting their enhanced capacities of energy dissipation. This behavior may be related to the better MCC dispersion in these composites. The MCC treatment with C_4_MImCl was only effective when the corresponding filler was used below its saturation in the matrix (below 10 wt.%), when the hydrogen bonding formation mechanism between the IL and hardener is more effective. In other words, it is possible that a high content of filler generates a high content of aggregates which, in turn, hinders the IL-hardener hydrogen bonding formation, as aforementioned

Figure 9 presents the damping curves, and Table 7 summarizes the main results from these curves. E/MCC_C_4_5 presented the highest T_g_. If a stronger interfacial region is formed between a polymer matrix and reinforcement, the T_g_ of the composite can increase [44]. Moreover, the shifting of the damping curves to higher temperatures indicates lower polymer chains mobility and stronger interfacial adhesion [54]. Furthermore, it is possible that the hydrogen bonding formation between the IL and AH-260 hardener, and, consequently, with the cured epoxy resin, promoted an enhanced interface, which decreased the polymer chain mobility at higher temperatures. This greater stiffness reduced its energy absorption capacity, related to the lower peak height in its damping curve, as better discussed in the toughness behavior section.

Despite the weaker hydrogen bond formed between C_2_OHMImCl and AH260 hardener (see Appendix A), compared to the other IL, there was a high interaction (possibly by hydrogen bonds) between MCC and C_2_OHMImCl as discussed in the XRD section. These results may be related to the highest peak of E/MCC_C_2_OH10 and to the higher values for tensile modulus and flexural strength, presented in Table 3 and Table 4, respectively.

The parameter FWHM may be used to assess composite homogeneity, and the lower the value, the higher the homogeneity (see Table 7) [37]. E/MCC10 presented the highest value, which can be related to the insufficient interfacial strength between MCC and epoxy to disperse homogeneously the high content of 10 wt.% of MCC.

A Cole–Cole plot can also be used to assess composite homogeneity [54,55]. When a better distribution of filler in the polymer matrix is reached, a smaller difference in relaxation times for each composite component is presented, narrowing the curve [8,54]. In comparison to E, all composites presented a broader curve (Figure 10). Moreover, due to their better results for E’ and E”, E/MCC_C_4_5, E/MCC_(HO_2_)_2_10 and E/MCC_C_2_OH10 presented curves with higher peaks, although the peaks of E/MCC_(HO_2_)_2_10 and E/MCC_C_2_OH10 were narrower. These results indicate that despite the differences in hydrogen bond strength between IL and matrix, the type of IL does not influence the MCC distribution into the composite, since there was a narrower Cole–Cole plot for all composites with MCC-IL.

## 4. Conclusions

The surface treatment of MCC with a low content of 1 wt.% of IL is reported as an alternative method for the modification of the filler surface. The use of such salts increased MCC crystallinity and thermal stability, which was most pronounced with IL C_2_OHMImCl. A better crystals’ dispersion as well as a rougher surface were also reported when IL was employed, regardless of the IL-type.

For the epoxy-based composites, the composites manufactured with 10 wt.% presented better mechanical, viscoelastic and toughness properties when unmodified-MCC was employed as reinforcement. However, when MCC-IL was used, those treated with C_2_OHMImCl presented higher quasi-static mechanical properties, perhaps related to the greater crystallinity showed in the XRD analysis, which is related to the high affinity between the reinforcement and IL. Furthermore, the formation of a strong hydrogen bonding between epoxy hardener-IL and MCC may favor the higher mechanical response of such composites.

In the same way, C_4_MImCl presented a stronger hydrogen bonding with AR-260 hardener, compared to the other IL, which was mainly responsible for the improvements on dynamical-mechanical properties of the composites, followed by those (HO_2_CC1)_2_MImCl, HO_2_CC1MImCl and C_2_OHMImCl, respectively, the latter with a better interaction with MCC compared to the other salts.

## Figures and Tables

**Figure 1 polymers-15-00333-f001:**
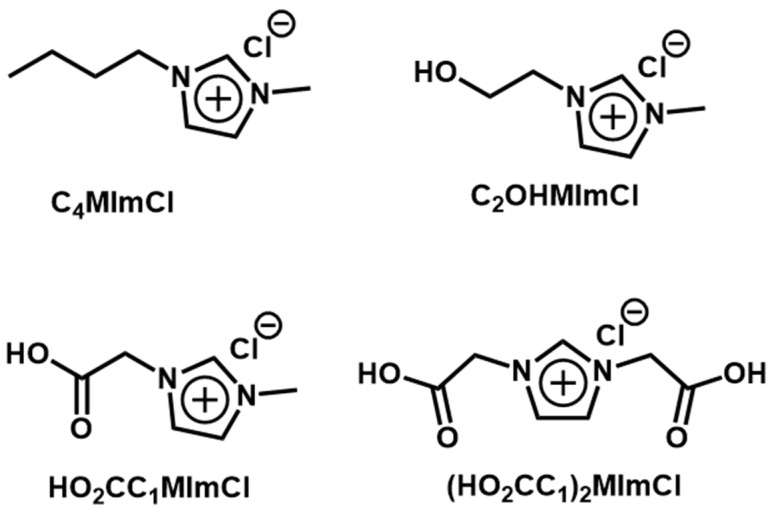
Chemical structures of the IL investigated as compatibilizers for MCC/epoxy composites.

**Figure 2 polymers-15-00333-f002:**
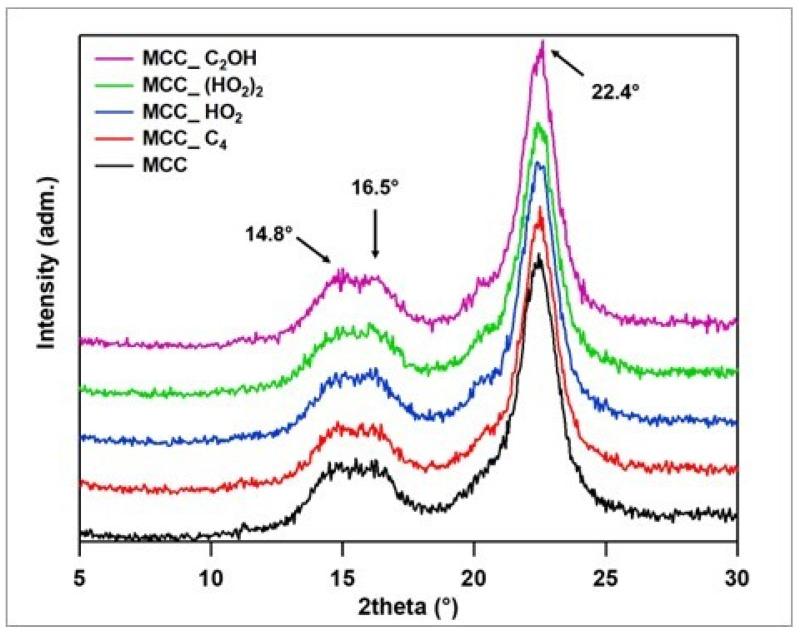
Diffractograms for the unmodified and IL-modified MCC.

**Figure 3 polymers-15-00333-f003:**
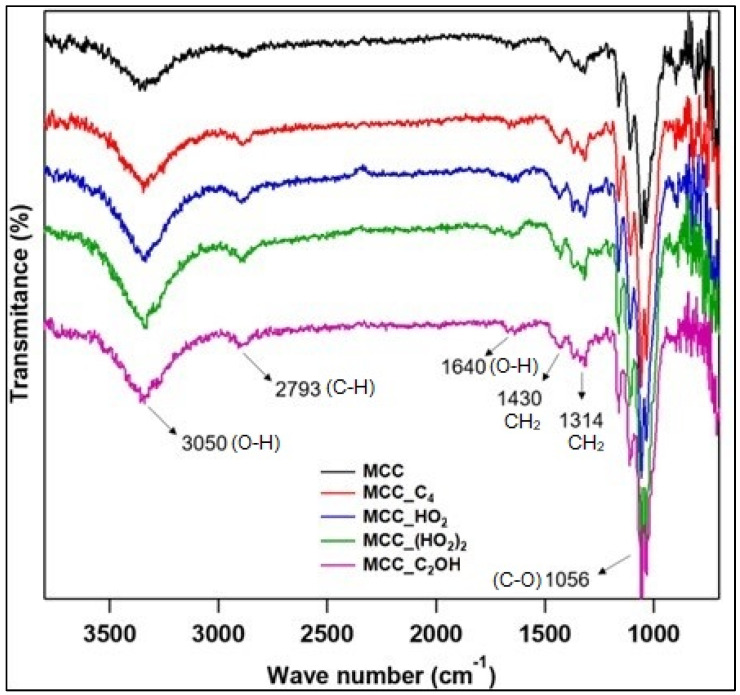
FTIR transmittance spectra of the MCC treated with 1 wt.% of the ILs.

**Figure 4 polymers-15-00333-f004:**
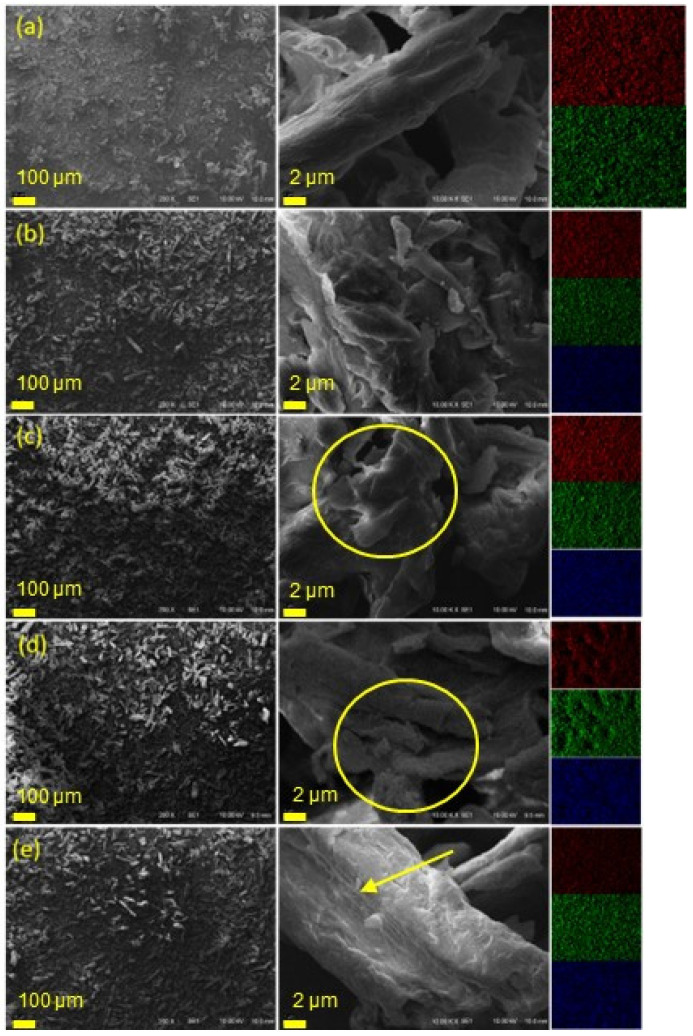
SEM micrographs and the respective EDS patterns (carbon = red, oxygen = green, chloride = blue) of (**a**) MCC, (**b**) MCC_C_4_, (**c**) MCC_HO_2_, (**d**) MCC_(HO_2_)_2_ and (**e**) MCC_C_2_OH.

**Figure 5 polymers-15-00333-f005:**
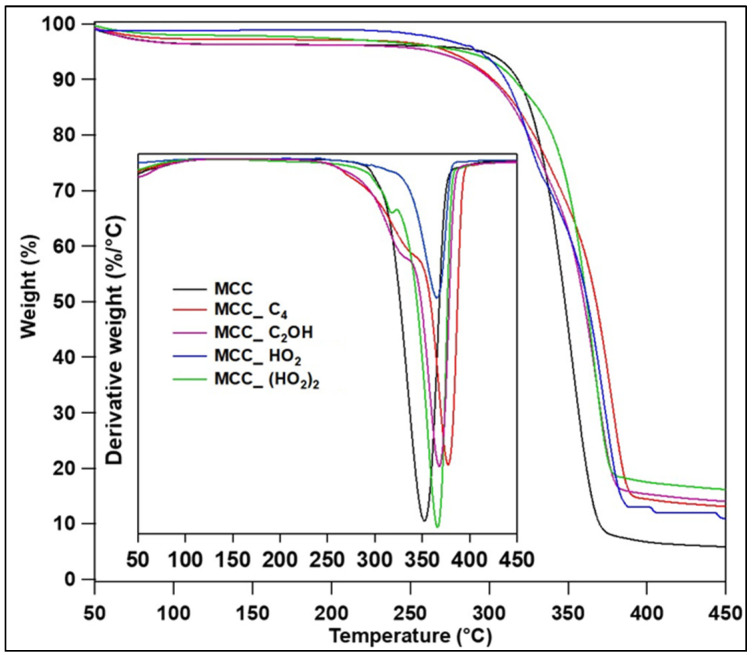
TG and DTG curves for the IL-treated MCC.

**Figure 6 polymers-15-00333-f006:**
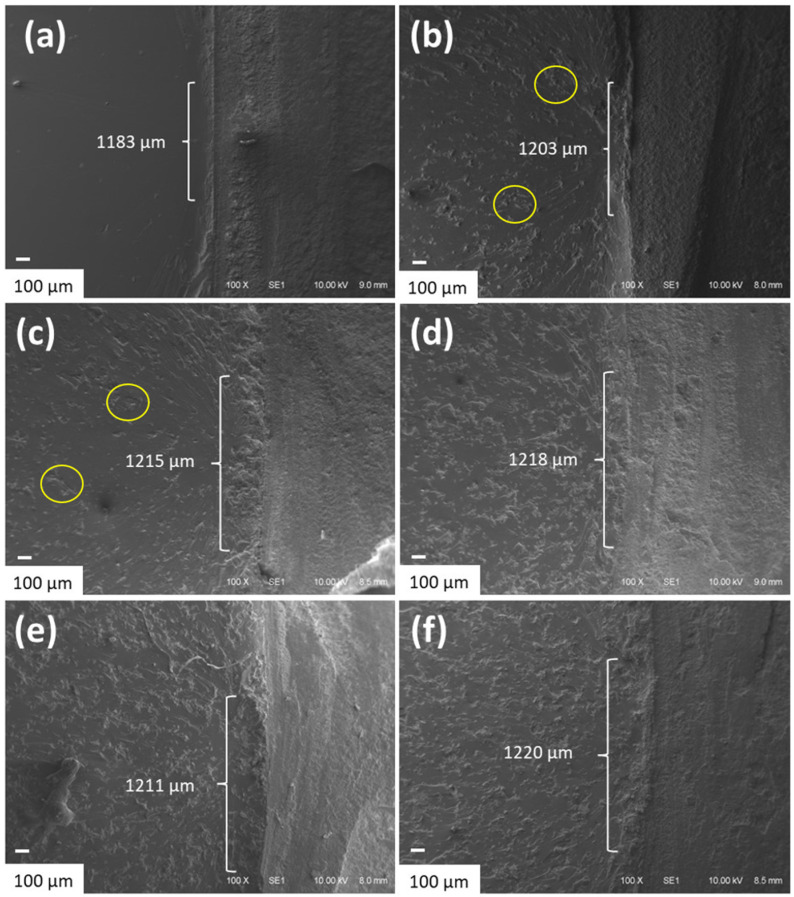
SEM micrographs of the crack’s initiation on the surface for neat epoxy (**a**), E/MCC5 (**b**), E/MCC_C_4_5 (**c**), E/MCC_C_4_10 (**d**), E/MCC_(HO_2_)_2_10 (**e**), and E/MCC_C_2_OH10 (**f**) (100×).

**Figure 7 polymers-15-00333-f007:**
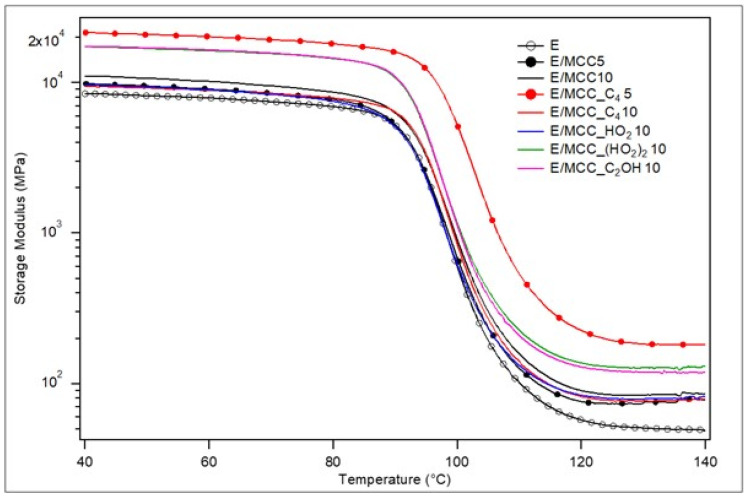
Storage modulus curves for epoxy and the MCC/epoxy composites.

**Figure 8 polymers-15-00333-f008:**
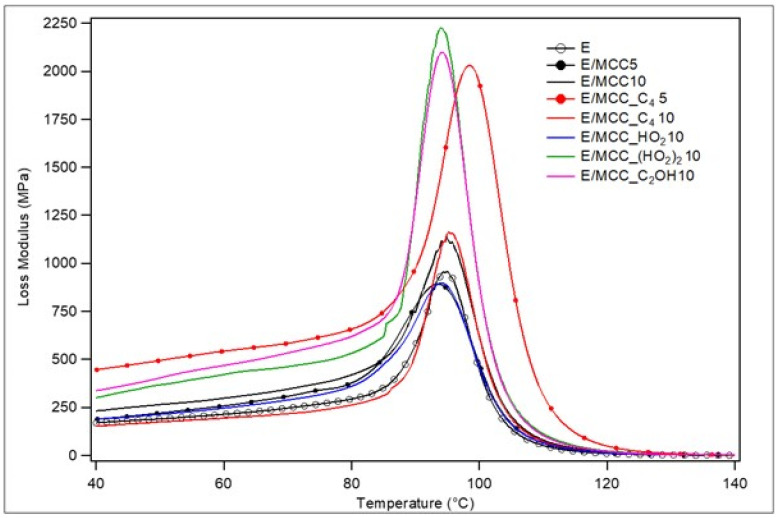
Loss modulus curves for epoxy and the MCC/epoxy composites.

**Figure 9 polymers-15-00333-f009:**
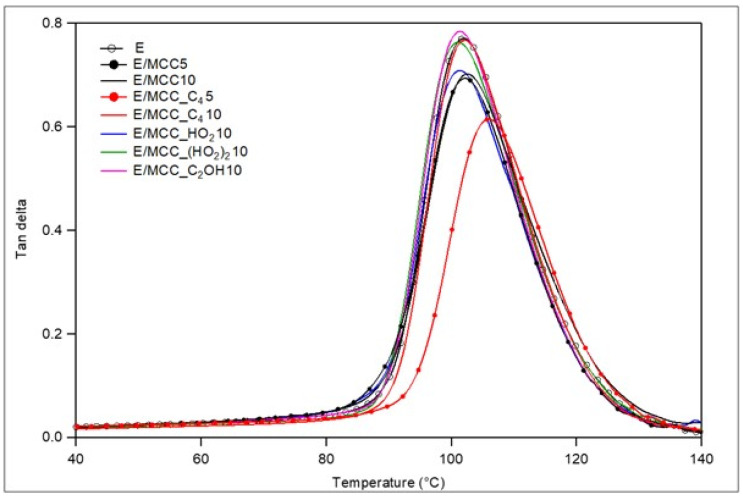
Damping curves for epoxy and the MCC/epoxy composites.

**Figure 10 polymers-15-00333-f010:**
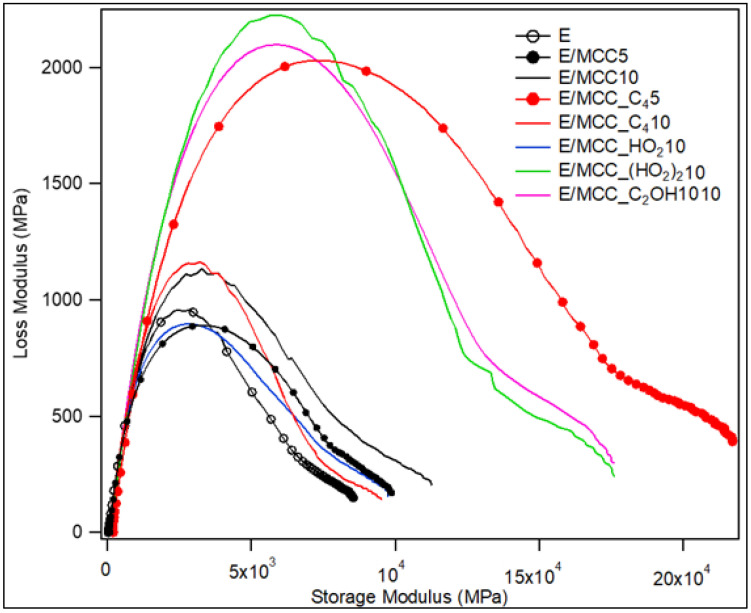
Cole-Cole curves for epoxy and the MCC/epoxy composites.

**Table 1 polymers-15-00333-t001:** Summary of the thermal decomposition data of the MCC samples, determined by TGA.

Sample	T_onset_ (°C)	T_max_ (°C)	Weight Loss at T_max_ (%)	Weight Loss at 374 °C (%)
MCC MCC_C_4_ MCC_C_2_OH MCC_HO_2_ MCC_(HO_2_)_2_	322.9 302.6 301.7 329.5 332.6	352.3 377.3 367.8 365.1 366.0	61.5 68.5 65.9 62.5 61.3	93.1 62.1 77.5 72.3 78.4

**Table 2 polymers-15-00333-t002:** DSC-derived curing data for epoxy and the MCC/epoxy composites.

Sample	T_onset_ (°C)	T_peak_ (°C)	Heat of Reaction (J/g)
E E/MCC10 E/MCC_C_4_10 E/MCC_HO_2_10 E/MCC_(HO_2_)_2_10 E/MCC_C_2_OH10	62.0 61.6 61.2 61.4 61.1 61.3	110.6 107.8 111.3 106.7 111.7 111.0	427.5 459.3 418.9 410.4 413.7 397.2

**Table 3 polymers-15-00333-t003:** Tensile test-derived results for epoxy and the MCC/epoxy composites.

Sample	Tensile Modulus (MPa) ^1^	Tensile Strength (MPa) ^1^	Poisson’s Ratio (ν) ^1^	Shear Modulus (MPa) ^1^
E E/MCC5 E/MCC10 E/MCC_C_4_5 E/MCC_C_4_10 E/MCC_HO_2_10 E/MCC_(HO_2_)_2_10 E/MCC_C_2_OH10	2696 ^3.5^ AB 2894 ^11.3^ BC 2487 ^10.8^ A 2608 ^6.1^ AB 3139 ^7.4^ CD 3285 ^6.8^ DE 2758 ^9.3^ AB 3535 ^10.9^ E	27.4 ^8.7^ ABC 44.5 ^6.4^ G 43.5 ^5.9^ FG 33.8 ^9.9^ CD 28.8 ^11.0^ B 37.6 ^13.5^ DE 24.5 ^9.9^ A 39.8 ^12.7^ EF	0.26 ^11.9^ D 0.21 ^9.8^ C 0.16 ^1.2^ B 0.18 ^2.3^ BC 0.21 ^4.4^ BC 0.19 ^2.4^ BC 0.14 ^3.6^ B 0.12 ^2.2^ A	1827.4 ^9.3^ BC 1952.6 ^11.9^ C 1498.0 ^11.0^ A 1615.2 ^5.3^ AB 1879.4 ^14.9^ BC 2018.9 ^11.7^ CD 1999.5 ^14.4^ CD 2274.7 ^13.6^ D

^1^ Values in superscript are the coefficient of variance; different letters represent statistical differences among the means for that property (*p* < 0.05).

**Table 4 polymers-15-00333-t004:** Flexural test-derived results for epoxy and the MCC/epoxy composites.

Sample	Flexural Modulus (MPa) ^1^	Flexural Strength (MPa) ^1^	Strain at Flexural Break (%) ^1^
E E/MCC5 E/MCC10 E/MCC_C_4_5 E/MCC_C_4_10 E/MCC_HO_2_10 E/MCC_(HO_2_)_2_10 E/MCC_C_2_OH10	2287 ^11.9^ A 2313 ^13.5^ A 2302 ^4.7^ A 2593 ^12.9^ AB 2723 ^6.7^ B 2734 ^13.7^ B 2502 ^13.2^ AB 2671 ^13.3^ B	65.6 ^10.7^ BC 73.8 ^11.2^ CD 50.9 ^9.7^ A 84.6 ^12.9^ E 76.3 ^8.6^ DE 63.1 ^13.8^ B 64.6 ^5.1^ B 67.3 ^12.8^ BC	11.6 ^5.6^ D 3.7 ^13.5^ BC 2.7 ^13.2^ A 4.3 ^13.0^ C 3.4 ^8.3^ B 2.5 ^13.7^ A 2.7 ^12.0^ A 2.5 ^14.9^ A

^1^ Values in superscript are the coefficient of variance; different letters represent statistical differences among the means for that property (*p* < 0.05).

**Table 5 polymers-15-00333-t005:** Fracture toughness (KIC) and strain energy release rate (GIC) for epoxy and the MCC/epoxy composites.

Sample	K_IC_ (MPa.m^−2^) ^1^	G_IC_ (kJ.m^−2^) ^1^
E E/MCC5 E/MCC10 E/MCC_C_4_5 E/MCC_C_4_10 E/MCC_HO_2_10 E/MCC_(HO_2_)_2_10 E/MCC_C_2_OH10	1.57 ^12.0^ B 1.32 ^13.4^ A 1.40 ^5.6^ AB 1.43 ^17.6^ AB 1.39 ^11.9^ AB 1.37 ^9.38^ AB 1.30 ^13.6^ A 1.30 ^13.1^ A	8.16 ^10.4^ AB 8.95 ^4.7^ AB 8.57 ^7.7^ AB 8.92 ^10.2^ AB 10.03 ^13.2^ C 9.23 ^11.7^ BC 7.99 ^8.6^ A 9.05 ^4.3^ B

^1^ Values in superscript are the coefficient of variance; different letters represent statistical differences among the means for that property (*p* < 0.05).

**Table 6 polymers-15-00333-t006:** Normalized storage moduli and effectiveness of fillers reinforcement (C) for the MCC/epoxy composites.

Sample	E_c_/E_m_ (at 40 °C)	E_c_/E_m_ (at 140 °C)	C (40/140 °C)
E/MCC5 E/MCC10 E/MCC_C_4_5 E/MCC_C_4_10 E/MCC_HO_2_10 E/MCC_(HO_2_)_2_10 E/MCC_C_2_OH10	1.16 1.31 2.55 1.12 1.15 2.05 2.07	1.59 1.74 3.69 1.60 1.67 2.65 2.41	0.73 0.75 0.69 0.70 0.68 0.78 0.86

**Table 7 polymers-15-00333-t007:** Damping curves-derived results for epoxy and the MCC/epoxy composites.

Sample	T_g_ (°C) ^1^	Peak Height ^1^	FWHM (°C) ^1^
E E/MCC5 E/MCC10 E/MCC_C_4_5 E/MCC_C_4_10 E/MCC_HO_2_10 E/MCC_(HO_2_)_2_10 E/MCC_C_2_OH10	101.88 101.63 102.42 106.51 101.86 101.20 100.88 101.38	0.76 0.69 0.69 0.61 0.76 0.70 0.76 0.78	17.6 18.3 19.33 17.48 17.31 18.28 18.67 17.51

^1^ Tg = glass transition temperature determined as the temperature at the peak of damping curves; FWHM = full width at half-maximum.

## Data Availability

Not applicable.

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
