# Peer review of "Imidazolium Ionic Liquids as Compatibilizer Agents for Microcrystalline Cellulose/Epoxy Composites"

_polymers, 2023, doi:10.3390/polym15020333_

Round 1

Reviewer 1 Report

In this manuscript, the authors employed a series of imidazolium ionic liquids as compatibilizer agents to improve the mechanical properties of micro-crystalline cellulose/epoxy composites. And the results illustrated that the aforementioned effects were associated to stronger hydrogen bonding between IL and epoxy hardener or the IL with MCC. These can supply some valuable information for the researchers to design sustainable epoxy composite. Based on its present format, this manuscript can be published in Polymers after the following items are clarified.

1.       In Introduction part, the authors introduced some strategies to overcome the fragility of epoxy resin, however, some recent references on new toughening methods needed to be supplemented. Such as, Macromolecules 2022, 55, 95029512, and Macromolecules 2021, 54, 77967807.

2.       The authors discussed the effect of addition of IL on the thermal stability (TGA data) of MCC, but did not mentioned the differences on epoxy/MCC composite between with and without IL, can the authors make any comment on this?

Author Response

In this manuscript, the authors employed a series of imidazolium ionic liquids as compatibilizer agents to improve the mechanical properties of micro-crystalline cellulose/epoxy composites. And the results illustrated that the aforementioned effects were associated to stronger hydrogen bonding between IL and epoxy hardener or the IL with MCC. These can supply some valuable information for the researchers to design sustainable epoxy composite. Based on its present format, this manuscript can be published in Polymers after the following items are clarified.

Comment: Thank you for your relevant comment and the time dedicated to deep evaluate and improve the quality of our manuscript. We separated your relevant comments and a brief comment for each modification is addressed bellow. The modifications in the main text are highlighted in yellow color, for a better visualization

  1. In Introduction part, the authors introduced some strategies to overcome the fragility of epoxy resin, however, some recent references on new toughening methods needed to be supplemented. Such as, Macromolecules 2022, 55, 9502−9512, and Macromolecules 2021, 54, 7796−7807.

Comment: We included the new methodologies suggested by you.

  1. The authors discussed the effect of addition of IL on the thermal stability (TGA data) of MCC, but did not mention the differences on epoxy/MCC composite between with and without IL, can the authors make any comment on this?

Comment: Thanks for your question. We performed the thermal stability analysis aiming to correlate the main changes in thermal stability with structural and chemical issues, when IL were employed in MCC. We performed it, aiming to verify if IL induced some relevant decrease in thermal stability, which is in part related to structural changes. In previous work, a significant change in thermal stability and degradation kinetic was reported, when IL was used for MCC modification in higher concentrations (doi:10.1016/J.CARPTA.2022.100211), so based on TGA, for these IL we did not verify any significant changes. Moreover, we did not perform thermal analysis for the composites, because this was not the main focus of the work, we focused on the main changes in thermo-mechanical response and toughness characteristics. We included a comment at the final of objectives, in the last paragraph of introduction section.

Reviewer 2 Report

1.      An interesting work, the writing can be improved, moreover, some figures are cropped and should be added in high quality.

2.      The introduction of IL to polymeric material is based on polarity, what are your comments on choosing chloride-based IL for epoxy/MCC?

3.      The morphology should be discussed more within context of composite, much detail was given for MCC only.

4.      Alone IL would dissolve or partially alters the MCC, can authors add comments on FTIR in detail for change on MCC surface is due to addition of functional groups, or else?

5.      Please comments on the interaction of IL with composites including MCC/epoxy, following can be used as references to discuss the interaction of imidazolium IL with MCC and polymeric materials.

a.      Effects of ionic liquid on cellulosic nanofiller filled natural rubber bionanocomposites

b.     Influence of ionic liquid on rheological behaviors of candle soot and cellulose nanocrystal filled natural rubber nanocomposites

c.      Large amplitude oscillatory rheology of silica and cellulose nanocrystals filled natural rubber compounds

6.      The conclusion part should be enhanced. Good work, best wishes.

Author Response

  1. An interesting work, the writing can be improved, moreover, some figures are cropped and should be added in high quality.

Comment: Thank you for your relevant comment and the time dedicated to deep evaluate and improve the quality of our manuscript. We separated your relevant comments in topics and they are presented below. We performed a deep revision of the manuscript aiming to find any mistakes and improve the English. The modifications in the main text are highlighted in yellow color, for a better visualization. We changed some Figures’ size and disposition aiming to improve the quality.

  1. The introduction of IL to polymeric material is based on polarity, what are your comments on choosing chloride-based IL for epoxy/MCC?

Comment: We chosen the IL based in chloride anion due to previous results (doi:10.1016/j.polymer.2021.123787). It was proved in previous report that IL with chloride anion presented a stronger hydrogen bonding, when compared to other anion-based IL, such as acetate. In future research we will evaluate the influence of other anions used for the improvement of compatibility of fillers into epoxy-based composites. Specifically in this one, we investigated the influence of the cation for the improvement of the compatibility of such reinforcements. We included a sentence at the final of the objectives elucidating this issue.

  1. The morphology should be discussed more within context of composite; much detail was given for MCC only.

Comment: We improved the discussion about the morphology of the composites with the IL-modified MCC, especially in the toughness section.

  1. Alone IL would dissolve or partially alters the MCC, can authors add comments on FTIR in detail for change on MCC surface is due to addition of functional groups, or else?

Comment: We performed the MCC-IL modification, following our methodology, that is based in previous work (doi:10.1002/app.48702) (doi:10.1002/app.46693) (doi:10.1007/s00289-018-2550-4). In previous works, we proved that the IL is adsorbed in the surface of fillers, when previously dissolved in an ethanolic solution. Aiming to prove the IL on the MCC surface, we performed EDS measurements, presented with the MCC SEM images (Fig. 4). Due to the fact that IL is physisorbed in the MCC surface, there was no changes of the main peaks of MCC, in the FTIR spectra. Furthermore, it is important to stress that the content of IL onto the MCC is low (1wt.%) which also does not enable the measurement of the salt (we included it in the discussion of FTIR results). As an additional comment, in previous work (https://doi.org/10.1016/j.carpta.2022.100211), when IL is used in higher content, peaks related to the imidazolium ring appears (close to 1562 cm-1).

  1. Please comments on the interaction of IL with composites including MCC/epoxy, following can be used as references to discuss the interaction of imidazolium IL with MCC and polymeric materials.
    a. Effects of ionic liquid on cellulosic nanofiller filled natural rubber bio nanocomposites
    b. Influence of ionic liquid on rheological behaviors of candle soot and cellulose nanocrystal filled natural rubber nanocomposites
    c. Large amplitude oscillatory rheology of silica and cellulose nanocrystals filled natural rubber compounds

Comment: We improved the discussion about the interactions of MCC-IL and epoxy, based in your relevant references. We included all the cited references in the main text.

  1. The conclusion part should be enhanced. Good work, best wishes.

Comment: We improved the conclusion section including more comments, based in the results. Thank you.

Reviewer 3 Report

The manuscript under the title: “Imidazolium ionic liquids as compatibilizer agents for microcrystalline cellulose/epoxy composites” is in line with Polymers journal. This topic is relevant and will be of interest to the readers of the journal. It based on original research. This research has scientific novelty and practical significance. The article has a typical organization for research articles.
Before the publication it requires significant improvements, especially:

1.     The "Introduction" section: To reduce the brittleness of epoxy composites, various plasticizers and fillers (including nanosized ones) are used. It is necessary to consider concrete examples. In addition, functionalization of fillers is very effective. I believe that it is necessary to add specific examples in the "Introduction" section and indicate their advantages and disadvantages. I think the related references should be cited corresponding to each aspect, e.g. (but not limited to these), which will undoubtedly improve the "Introduction" section:

- Russ J Appl Chem 86, 765–771 (2013). https://doi.org/10.1134/S107042721305025X

- Polymers 2022, 14(23), 5064; https://doi.org/10.3390/polym14235064

Polymers 2022, 14(2), 338; https://doi.org/10.3390/polym14020338

Polymer Composites, 36, 1891-1898 (2015). https://doi.org/10.1002/pc.23097

Appl. Polym. Sci. 2019, 136, 47410, https://doi.org/10.1002/app.47410

  1. Section 2. It is necessary to add the physicochemical characteristics of components - give a table with the main physicochemical and technological properties of epoxy resin, hardener, MCC and IL.
  2. Line 211-213. This text must be deleted, because this is the recommendation text from the template.
  3. All functional groups (О-Н, С-Н, etc.) must be shown on the spectra.
  4. Table 1. In my opinion, it is necessary to fix the final weight loss not at 374 C for all samples, but for each sample individually at the temperature when the weight loss curve reaches a plateau and, therefore, the sample weight loss ends. Otherwise, you will get contradictory data.
  5. How do the authors explain the decrease in tensile strength with the introduction of modified MCC, while the bending strength increases?
  6. Line 360-361. It would be good in this case to cite not your previous work, but the work of other scientific groups, I recommend you cite the following work:

-        Polymers 2020, 12(7), 1437; https://doi.org/10.3390/polym12071437

-        Composites Part B: Engineering 2017, 114, 175-183; https://doi.org/10.1016/j.compositesb.2017.01.032

8.     Line 417-420. These conclusions must be supported by appropriate references to the literature.

Author Response

The manuscript under the title: “Imidazolium ionic liquids as compatibilizer agents for microcrystalline cellulose/epoxy composites” is in line with Polymers journal. This topic is relevant and will be of interest to the readers of the journal. It based on original research. This research has scientific novelty and practical significance. The article has a typical organization for research articles.
Before the publication it requires significant improvements, especially:

Comment: Thank you for your relevant comment and the time dedicated to deep evaluate and improve the quality of our manuscript. We separated your relevant comments in topics and a brief comment for each modification is addressed bellow. The modifications in the main text are highlighted in yellow color, for a better visualization

  1. The "Introduction" section: To reduce the brittleness of epoxy composites, various plasticizers and fillers (including nanosized ones) are used. It is necessary to consider concrete examples. In addition, functionalization of fillers is very effective. I believe that it is necessary to add specific examples in the "Introduction" section and indicate their advantages and disadvantages. I think the related references should be cited corresponding to each aspect, e.g. (but not limited to these), which will undoubtedly improve the "Introduction" section:
    - Russ J Appl Chem 86, 765–771 (2013). https://doi.org/10.1134/S107042721305025X
    - Polymers 2022, 14(23), 5064; https://doi.org/10.3390/polym14235064
    - Polymers 2022, 14(2), 338; https://doi.org/10.3390/polym14020338
    - Polymer Composites, 36, 1891-1898 (2015). https://doi.org/10.1002/pc.23097
    - Appl. Polym. Sci. 2019, 136, 47410, https://doi.org/10.1002/app.47410

Comment: We included a paragraph about the use of different fillers for the improvement of properties of epoxy. We also enhanced the section about the functionalization of such fillers.

  1. Section 2. It is necessary to add the physicochemical characteristics of components - give a table with the main physicochemical and technological properties of epoxy resin, hardener, MCC and IL.

Comment: We included some physicochemical characteristics of the components required in the main text. These reactants were used as received.

  1. Line 211-213. This text must be deleted, because this is the recommendation text from the template.

Comment: We removed the sentence, thank you to pay attention on that.

  1. All functional groups (О-Н, С-Н, etc.) must be shown on the spectra.

Comment: We included the functional groups in its respective transmittance bands, in the FTIR spectra (Fig. 3).

  1. Table 1. In my opinion, it is necessary to fix the final weight loss not at 374 C for all samples, but for each sample individually at the temperature when the weight loss curve reaches a plateau and, therefore, the sample weight loss ends. Otherwise, you will get contradictory data.

Comment: We chosen to represent the weight loss at 374 C, because in this temperature occurs mainly the thermal decomposition of cellulose. Regarding literature, this temperature is used for the analysis of modified MCC, because the main thermal decomposition occurs. So, its analysis is essential in addition to the evaluation of Tmax (the temperature on the plateau, as recommended by you) , also reported. As an additional comment, we evaluated many weight loss and temperature throw the TGA analysis and discuss it taking into account many issues, as included in the main text.

  1. How do the authors explain the decrease in tensile strength with the introduction of modified MCC, while the bending strength increases?

Comment: We included a sentence in the main text. Briefly, differently from tensile tests, the flexural behavior of composites is lower dependent on the failure type and samples preparation. Irregular surfaces and imperfections may cause premature failure of materials, in tensile. Then, lower standard deviation and a better accuracy on the measuring of failure is evident in flexural tests.

  1. Line 360-361. It would be good in this case to cite not your previous work, but the work of other scientific groups, I recommend you cite the following work:
    - Polymers 2020, 12(7), 1437; https://doi.org/10.3390/polym12071437
    - Composites Part B: Engineering 2017, 114, 175-183; https://doi.org/10.1016/j.compositesb.2017.01.032

Comment: We included the suggested references for the discussion of such result.

  1. Line 417-420. These conclusions must be supported by appropriate references to the literature.

Comment: We included a reference that supports such statement.

Round 2

Reviewer 2 Report

No more comments to add.

Reviewer 3 Report

The authors considered most of the comments or adequately responded to the remarks contained in the review; therefore, the work may be approved for publication.